# Sphingolipid Profile during Cotton Fiber Growth Revealed That a Phytoceramide Containing Hydroxylated and Saturated VLCFA Is Important for Fiber Cell Elongation

**DOI:** 10.3390/biom11091352

**Published:** 2021-09-12

**Authors:** Qian Chen, Fan Xu, Li Wang, Xiaodong Suo, Qiaoling Wang, Qian Meng, Li Huang, Caixia Ma, Guiming Li, Ming Luo

**Affiliations:** 1Key Laboratory of Biotechnology and Crop Quality Improvement, Ministry of Agriculture/Biotechnology Research Center, Southwest University, Chongqing 400716, China; chenqiansuaige@163.com (Q.C.); xufanfeiren@163.com (F.X.); sxd18883770582@126.com (X.S.); wql19980513@163.com (Q.W.); mqhongbin@foxmail.com (Q.M.); hl19970914@email.swu.edu.cn (L.H.); mcx2116@email.swu.edu.cn (C.M.); lgm5683@163.com (G.L.); 2Key Laboratory of Horticulture Science for Southern Mountains Regions of Ministry of Education, College of Horticulture and Landscape Architecture, Southwest University, Chongqing 400716, China; 3State Cultivation Base of Crop Stress Biology for Southern Mountainous Land, Academy of Agricultural Sciences, Southwest University, Chongqing 400716, China; 4Zhengzhou Research Base, State Key Laboratory of Cotton Biology, Zhengzhou University, Zhengzhou 450001, China; wangli07-2@163.com; 5State Key Laboratory of Cotton Biology, Institute of Cotton Research, Chinese Academy of Agricultural Sciences, Anyang 455000, China

**Keywords:** sphingolipid, VLCFA, cotton, fiber cell, elongation, secondary cell wall

## Abstract

Cotton fiber is a single-celled seed trichrome that arises from the epidermis of the ovule’s outer integument. The fiber cell displays high polar expansion and thickens but not is disrupted by cell division. Therefore, it is an ideal model for studying the growth and development of plant cells. Sphingolipids are important components of membranes and are also active molecules in cells. However, the sphingolipid profile during fiber growth and the differences in sphingolipid metabolism at different developmental stages are still unclear. In this study, we detected that there were 6 classes and 95 molecular species of sphingolipids in cotton fibers by ultrahigh performance liquid chromatography-MS/MS (UHPLC-MS/MS). Among these, the phytoceramides (PhytoCer) contained the most molecular species, and the PhytoCer content was highest, while that of sphingosine-1-phosphate (S1P) was the lowest. The content of PhytoCer, phytoceramides with hydroxylated fatty acyls (PhytoCer-OHFA), phyto-glucosylceramides (Phyto-GluCer), and glycosyl-inositol-phospho-ceramides (GIPC) was higher than that of other classes in fiber cells. With the development of fiber cells, phytosphingosine-1-phosphate (t-S1P) and PhytoCer changed greatly. The sphingolipid molecular species Ceramide (Cer) d18:1/26:1, PhytoCer t18:1/26:0, PhytoCer t18:0/26:0, PhytoCer t18:1/h20:0, PhytoCer t18:1/h26:0, PhytoCer t18:0/h26:0, and GIPC t18:0/h16:0 were significantly enriched in 10-DPA fiber cells while Cer d18:1/20:0, Cer d18:1/22:0, and GIPC t18:0/h18:0 were significantly enriched in 20-DPA fiber cells, indicating that unsaturated PhytoCer containing hydroxylated and saturated very long chain fatty acids (VLCFA) play some role in fiber cell elongation. Consistent with the content analysis results, the related genes involved in long chain base (LCB) hydroxylation and unsaturation as well as VLCFA synthesis and hydroxylation were highly expressed in rapidly elongating fiber cells. Furthermore, the exogenous application of a potent inhibitor of serine palmitoyltransferase, myriocin, severely blocked fiber cell elongation, and the exogenous application of sphingosine antagonized the inhibition of myriocin for fiber elongation. Taking these points together, we concluded that sphingolipids play crucial roles in fiber cell elongation and SCW deposition. This provides a new perspective for further studies on the regulatory mechanism of the growth and development of cotton fiber cells.

## 1. Introduction

Cotton is an important natural fiber crop throughout the world. Cotton fiber is a single-celled fiber formed by the polar expansion and the secondary cell wall (SCW) thickening of the ovule epidermal cell. It is considered an ideal material for studying cell elongation, cellulose synthesis, and secondary wall deposition [1,2]. The development of fiber cells can be divided into four distinct and overlapping stages: initiation, elongation (primary cell wall formation), SCW deposition, and dehydration. On the day of anthesis, a large number of fiber cells are initiated from the ovule’s surface, followed by polar elongation, which is accompanied by the formation of the primary cell wall. At about 10 DPA (days post-anthesis), the elongation rate of the fiber reaches a peak (>2 mm/day), which is called the rapid fiber elongation stage [3,4,5]. Fiber cell elongation can last until 16–20 DPA. After about 15 DPA, fiber elongation gradually stops, and SCW synthesis begins. Therefore, the stage from 15 DPA to 20 DPA is also known as the transition period from primary cell wall synthesis to SCW synthesis, or the initiation period of SCW fiber synthesis [3]. After 20 DPA, fiber elongation stops completely, and the SCW is stably synthesized. The SCW deposition period lasts about 45 DPA, and the thickness of the fiber SCW finally reaches 3–4 μm [3]. The elongation and SCW synthesis stages are the two most important stages for fiber development in the determination of fiber quality (length, strength, and fineness). In the past two decades, extensive studies have focused on understanding the regulatory mechanisms underlying fiber development [6,7]. However, the molecular mechanism of cotton fiber cell elongation and SCW formation is still unclear.

Sphingolipids are complex lipids that are found in all animals, plants, fungi, and in a few prokaryotes and viruses [8]. The sphingolipids molecule consists of three main components: the long chain base (LCB) of sphingosine, the long chain fatty acids (LCFA) or the very long chain fatty acids (VLCFA), and the polar head group. [9,10,11]. Both LCBs and either LCFA or VLCFA can be further modified, such as by hydroxylation at the C-4 position of LCBs to form 3-hydroxy LCB or by desaturation at C4 and C8 of LCBs to form unsaturated LCBs [10], and LCFA or VLCFA can also be hydroxylated and desaturated [9,12]. Sphingolipid synthesis begins in the ER with the condensation of serine and palmitoyl-CoA by serine palmitoyltransferase (SPT) to produce 3-ketosphingosine. In the second step, 3-ketosphingosine is reduced to sphingosine by 3-ketosphinganine reductase (KSR). Sphingolipid structural complexity arises in the ceramide generated through the amide linkage of a fatty acid (FA) to an LCB. LCBs with dihydroxy LCBs are preferentially acylated with C16 FAs (16:0-CoA) by ceramide synthase I (CSI), which is primarily for glucosylceramide synthesis. Trihydroxy LCBs are preferentially acylated with very long chain fatty acyl (VLCFA)-CoAs by ceramide synthase II (CSII) for glucosylceramide (GluCer) and glycosyl-inositol-phospho-ceramide (GIPC) synthesis. In these processes, the LCBs can be desaturated by Δ4 desaturases (Δ4 DES) or Δ8 desaturases (Δ8 DES); the FAs can be hydroxylated by fatty acid α-hydroxylase (FA α-ohase). Sphingosines can be catalyzed by the LCB kinase (LCBK) to produce the sphingosine-1-phosphate (S1P), which can be dephosphorylated by the LCB-P phosphatase (LCB-P-Pase) to regenerate sphingosines. Ceramides can be catalyzed by ceramide kinase (CERK) to produce ceramide 1-phosphate (Cer-1-P), which can be dephosphorylated in the presence of ceramide phosphate phosphatase (Cer-P-Pase) to regenerate ceramides. Ceramides can also be reduced by ceramidase (CER) to form free LCBs [9,12,13]. Sphingolipids are major structural components of the plasma membrane, vacuole membrane, and inner membrane and are enriched in the lipid rafts of cell membranes [13,14,15]. Sphingolipids are not only a crucial component of biomembranes but are also an important bioactive molecule that mediates a variety of cell processes, such as programmed cell death [16,17,18,19], pathogen-induced hypersensitivity [18,20,21], the closure of stomatal guard cells regulated by ABA signaling [20,21,22,23], host–pathogen crosstalk [24], low temperature signal transduction [25,26,27], and the regulation of membrane stability [28].

Given that cotton fibers are extremely elongated single cells, it can be speculated that membranes might play crucial roles in the growth and development of cotton fiber [29,30]. Sphingolipids and sterols are important components of cell membranes and comprise the lipid raft of biomembranes, which play roles in the regulation of membrane fluidity and permeability, and the activity of membrane binding proteins [15,31,32]. Interestingly, a sphingolipid synthesis inhibitor, Fumonisin B1 (FB1), can modify the activity of membrane lipid rafts during the development of fibers and can then inhibit the elongation of cotton fiber [29,33]. Qi et al. reported that the content of saturated VLCFA in elongated fibers was far higher than that in ovules of the wild-type and the fuzzless-lintless mutant. The application of acephrachlor ACE, an inhibitor of fatty acid synthesis, can block the elongation of fibers [34]. Given that VLCFAs are predominantly present as sphingolipids, the results indicated that sphingolipids might play pivotal roles in the growth and development of fibers. However, the content and composition of sphingolipids in various developmental stages, especially in the stages of fiber cell elongation and SCW deposition, are still unclear.

Here, we characterized the content and composition of sphingolipids in fiber cells at four developmental stages by ultra-high performance liquid chromatography-MS/MS (UHPLC–MS/MS) and analyzed the differences in sphingolipid molecule species between the elongation stage and the SCW deposition stage of fiber cells. The results showed that the phytoceramide (PhytoCer) content was the highest, while that of sphingosine-1-phosphate (S1P) was the lowest. The sphingolipids PhytoCer, phytoceramides with hydroxylated fatty acyls (PhytoCer-OHFA), phyto-glucosylceramides (Phyto-Glucer), and glycosyl-inositol-phospho-ceramides (GIPC) were higher than the others in cotton fibers. The possible role of sphingolipids in the development of fiber cells was discussed. Our study provides a new perspective for further studies on the regulatory mechanism of cotton fiber cell growth and development.

## 2. Materials and Methods

### 2.1. Plant Materials and Growth Conditions

The cotton plant used in this study was upland cotton (*Gossypium hirsutum* L.) cv. Jimian14, which was kindly provided by professor Zhiying Ma (Hebei Agricultural University, China). All cotton plants were grown under natural field conditions in Chongqing.

### 2.2. Sample Labeling and Collection

In order to obtain seed samples at identical developmental stages, all of the flowers were labeled on the day of anthesis (0 DPA, day post-anthesis). The labeling was conducted every day and lasted for 30 days. After 30 days of labeling, cotton bolls were collected simultaneously at 5, 10, 15, 20, and 25 DPA.

### 2.3. In Vitro Ovule Culture and Fiber Length Measurement

For the in vitro ovule cultures, cotton ovules (*Gossypium hirsutum* L.) were collected at 2 DPA, sterilized in a 3‰ H_2_O_2_ solution, and cultured in Beasley and Ting’s medium [35] with 20 µM myriocin or 20 µM myriocin + 10 µM sphingosine at 32 °C in the dark for 10 days. BT medium adjusted with the amount of DMSO (dimethyl sulfoxide) equivalent to that used to dissolve myriocin and sphingosine was used as a blank control. To test the fiber length, at least 10 ovules were used. The cultured ovules were immersed in 30% glacial acetic acid and were heated in boiling water until the fibers dispersed. The ovules were placed on a slide and were rinsed with water to straighten out the fibers, which were then measured. Three biological replicates were performed in the experiment.

### 2.4. Lipid Extraction and Lipidomics

Fibers collected from cotton bolls at 5, 10, 15, and 20 DPA were placed in liquid nitrogen and were kept at −80 °C. After sample collection had been completed, lipid extraction and lipidomic analyses were performed by the Lipidall Technologies Company Limited (http://www.lipidall.com/ (accessed on 22 June 2019)), as previously described [11,33,36,37]. Briefly, the analyses were conducted using an Exion ultra-performance liquid chromatograph (UPLC) (AB Sciex, Redwood City, CA, USA) coupled with a Sciex QTRAP 6500 PLUS (AB Sciex, Redwood City, CA, USA). The lipids were separated using a Phenomenex Luna 3 µm silica column (Phenomenex, Torrance, CA, USA) (internal diameter: 150 × 2.0 mm) under the following conditions: mobile phase A (chloroform: methanol: ammonium hydroxide, 89.5:10:0.5) and mobile phase B (chloroform: methanol: ammonium hydroxide: water, 55:39:0.5:5.5). The gradient began with 95% of mobile phase A for 5 min and was followed by a linear reduction to 60% mobile phase A over 7 min. The gradient was held for 4 min, and mobile phase A was then further reduced to 30% and was held for 15 min. MRM transitions were constructed for a comparative analysis of the various sphingolipids. The individual sphingolipid classes were quantified by referencing spiked internal standards, namely Cer d18:1/17:0, GluCer d18:1/12:0, d17:1-S1P, D-ribo-phytosphingosine C17, and d17:1-Sph from Avanti Polar Lipids (Alabaster, AL, USA) and GM1 d18:1/18:0-d3 from Matreya LLC. (State College, PA, USA).

### 2.5. RNA Extraction and qRT-PCR

Total RNA from the 5, 10, 15, 20, and 25 DPA cotton fibers was extracted using the RNAprep pure Plant Kit (TIANGEN, Beijing, China). First-strand cDNA was synthesized using the PrimeScript RT reagent Kit with gDNA Eraser (TAKARA, Kyoto, Japan). qRT-PCR analysis was performed using Novostar-SYBR Supermix (Novoprotein, Shanghai, China): 94 °C for 2 min followed by 40 cycles of 94 °C for 30 s, 56 °C for 30 s, and 72 °C for 1 min. Three biological repetitions were performed. The specific primers of the selected genes and the internal control, HISTONE3 (GenBank accession No. AF024716), are listed in Appendix A.

### 2.6. Statistical Data Analysis

Orthogonal partial least squares discriminant analysis (OPLS-DA) is a modified PLS-DA (partial least squares discriminant analysis) method that can filter out the noise that is irrelevant to the classification information and can improve the analytical ability and effectiveness of a model. On the OPLS-DA score chart, there are two principal components: the predictive principal component and the orthogonal principal component. OPLS-DA maximizes the differences between groups on t1, so it can distinguish the variation directly from t1, while the variation in the orthogonal principal component reflects the variation in the group. The VIP is used to analyze the variables that have great influence on the model and is obtained according to the influence of the variables on the projection by SIMCA-P software 14.1. In this study, the OPLS-DA model was used to identify the difference in the sphingolipids between two fiber samples. Both R2Y and Q2 were the evaluation parameters of the model.

Data are presented as means ± SD. Statistical data analyses were performed by the one-tailed Student’s *t*-test, and *** indicates significant differences at *p* < 0.001.

## 3. Results

### 3.1. Sphingolipid Profile in Fiber Cells

In order to understand the changes in the sphingolipid composition and content following fiber cell development, we detected the sphingolipid profile of cotton fiber cells at four developmental stages by means of UHPLC–MS/MS. The results showed that (Figure 1A) 6 classes of sphingolipids and 95 molecular species of sphingolipids were detected (Appendix A), including ceramides (Cer), glucosylceramides (GluCer), phytoceramides (PhytoCer), GIPC, sphingosines (Sph), and sphingosine-1-phosphate (S1P); the number of molecular species was 15, 20, 35, 18, 3, and 4 for each class, respectively. The contents of phytosphingosines (PhytoSph), PhytoCer, PhytoCer-OHFA, Phyto-Glucer, and GIPC were higher, while the contents of Cer, GluCer, Sph, phytosphingosine-1-phosphate (t-S1P), and S1P were lower in the fiber cells (Figure 1B). The content of the sphingolipids containing hydroxylated and non-hydroxylated fatty acid chains was similar. The PhytoSph content was the highest in fiber cells at 5 DPA and gradually decreased with fiber growth. The content of t-S1P was lower in fiber cells at 5 DPA and gradually increased and then peaked at 15 DPA and finally decreased rapidly. The Cer content was the lowest at 5 and 10 DPA but was higher at 15 and 20 DPA. The GluCer content was higher in fiber cells at 5 and 15 DPA but was lower at 10 and 20 DPA. The changes in the other sphingolipid classes were moderate during cotton fiber development. These results suggest that during the development of fibers, different classes of sphingolipids showed different trends and that the contents of phytoSph, phyto-S1P, Cer, and GluCer varied greatly at different stages.

### 3.2. The Profile of Simple Sphingolipids in Fiber Cells

There are three Sph molecular species in fiber cells, including two phytoSph species and one Sph molecular species. The content of Sph t18:0 was the highest, followed by Sph t18:1, and Sph d18:1 was the lowest. The Sph t18:0 and Sph t18:1 content was the highest in the fiber cells at 5 DPA and gradually decreased along with fiber development (Figure 2A). The content of S1P t18:1 and S1P t18:0 was much higher than that of S1P d18:1 and S1P d18:0, indicating that S1P containing tri-hydroxyl LCB was the main S1P in fiber cells. S1P t18:1 and S1P t18:0 were the lowest at 5 DPA and gradually increased with fiber growth, reaching a peak at 15 DPA, then rapidly decreasing (Figure 2B). Given that the period from 5 DPA to 5 DPA is the elongation period for fiber cells, the results indicated that phyto-S1P is closely related to fiber elongation.

The content of Cers containing saturated LCB was lower than that of Cers containing unsaturated LCB. Among the Cers containing unsaturated LCB, the content of Cer containing saturated FA chains was higher than that containing unsaturated FA chains (except for Cer d18:1/20:1). The results (Figure 2C) indicated that the major Cer is composed of unsaturated LCB chains and saturated FA chains in fiber cells. During fiber development, the content of Cer (containing unsaturated LCB chains and saturated FA chains), such as the content of Cer d18:1/16:0, Cer d18:1/18:0, Cer d18:1/20:0, Cer d18:1/22:0, Cer d18:1/24:0, and Cer d18:1/26:0, was lower at 5 and 10 DPA but higher at 15 and 20 DPA in these cotton fibers. This indicated that Cers containing unsaturated LCB chains and saturated FA chains are associated with the inhibition of elongation and the promotion of SCW synthesis in cotton fibers. On the contrary, the content of Cers containing unsaturated LCBs and unsaturated FA, such as Cer d18:1/20:1, Cer d18:1/22:1, Cer d18:1/24:1, and Cer d18:1/26:1, was higher at 5 DPA (or 10 DPA) but decreased with fiber growth. These results indicated that the content of Cer containing unsaturated LCBs and saturated LCFA or VLCFA was high in fiber and their content was strikingly altered during the elongation and SCW synthesis stages of the fiber cells. It can be suggested that different molecular species of Cers might play different roles in fiber elongation and SCW formation.

PhytoCers contain tri-hydroxyl LCBs. In total, 35 PhytoCer molecular species were detected in fiber cells. There was no significant difference in content between the PhytoCer species containing hydroxylated FA and those containing non-hydroxylated FA (Figure 1B and Figure 2D). The PhytoCers containing VLCFA (in which the C atom number is even, such as C22, C24, and C26) were the predominant sphingolipids in the fiber cells. Among these, the PhytoCer species containing hydroxylated VLCFA were the highest at the 5 DPA stage; they then gradually decreased with the growth and development of fiber cells (Figure 2D), suggesting that these molecular species are conducive to fiber elongation. Three PhytoCer molecular species (t18:1/h26:1, t18:0/h22:1, and t18:0/h24:1) were much higher at the SCW deposition stage, indicating that they might play some role in SCW synthesis.

### 3.3. The Profile of Complex Sphingolipids in Fiber Cells

Complex sphingolipids include glucosylceramide (GluCer) and GIPC. In total, 20 GluCer molecular species were detected in the fiber cells, 12 of which were phyto-GluCers (Figure 3A). All of the detected GluCer molecular species contained hydroxylated FA and desaturated LCBs, and only one molecular species contained saturated LCBs. The phyto-GluCers containing saturated VLCFA were high in content, and phyto-GluCer t18:1/h24:0 had the highest content in the fiber cells. The GluCer molecules containing saturated C18 and C20 FAs were more numerous than the others, indicating that these GluCer molecules were predominant in fiber cells. During the growth and development of fiber cells, the content of the two most abundant molecular species of GluCers was higher at 5 and 15 DPA than at 10 and 20 DPA. All phyto-GluCers displayed a similar change in fiber growth. The highest content was present in fibers at 5 DPA, which then gradually decreased; the lowest content was found at 20 DPA.

In total, 18 GIPC molecular species were detected in the fiber cells. The GIPC with the highest content was t18:0/h26:0 followed by d18:0/h22:1 (or d18:1/h22:0), d18:0/h16:0, t18:0/h16:0, d18:0/h22:0, and t18:0/h18:0 (Figure 3B). Almost all of the GIPC molecules contained hydroxylated and saturated FA and saturated LCBs. During the growth and development of the fiber cells, the content of the GIPCs t18:0/h18:0, t18:0/h26:0, d18:0/h22:1, and d18:0/h22:0 was significantly higher in fibers at 15 and 20 DPA than in fibers at 5 and 10 DPA, indicating that most GIPCs were associated with SCW formation in fiber cells.

### 3.4. Differences of Sphingolipids between the Elongation and SCW Formation Stages

The elongation stage (the primary wall synthesis stage) and the SCW synthesis stage are two important stages in fiber cell development and have an important effect on the fiber quality and yield. The elongation period lasts about 16–20 days (from ~3 DPA to ~20 DPA). During this period, the rate and duration of elongation influences the final length of the fibers. The secondary wall synthesis period lasts about 20–25 days (from ~20 DPA to ~45 DPA). In this stage, the rate and duration of cellulose synthesis and secondary wall deposition affect the strength and fineness of mature fibers. As shown in the score plot (Figure 4A), there were six scores for each group, and two groups were clearly separated. In the model, R2Y = 0.997 and Q2 = 0.949 for the F-20 and F-10 groups (Figure 4A), and R2Y = 0.998 and Q2 = 0.962 for the F-5 and F-15 groups (Figure 5A), indicating that the quality of the OPLS-DA model was excellent for screening the key sphingolipid differences between the two samples. Compared to fibers at 10 DPA, three sphingolipid molecular species, Cer d18:1/20:0, Cer d18:1/22:0, and GIPC t18:0/h18:0, were significantly upregulated (*p*-value (*p*) < 0.05, fold change (FC) > 1.5), and seven sphingolipid molecular species, Cer d18:1/26:1, PhytoCer t18:1/ 26:0, PhytoCer t18:0/26:0, PhytoCer t18:1/h20:0, PhytoCer t18:1/ 26:0, PhytoCer t18:0/h26:0, and GIPC t18:0/h16:0, were significantly downregulated (*p* < 0.05, FC < 1/1.5) at 20 DPA (Figure 4B). The top five molecular species with the greatest amount of change were PhytoCer t18:1/26:0, PhytoCer t18:1/h26:0, PhytoCer t18:1/h20:0, GIPC t18:0/h18:0, and Cer d18:1/26:1, all of which accumulated greatly at the elongation stage, except for GIPC t18:0/h18:0 (Figure 4C). The results showed that Cers accumulated during the SCW deposition stage, while PhytoCers, especially unsaturated PhytoCers containing saturated and hydroxylated C26 FA, significantly accumulated during the fiber cell elongation stage.

### 3.5. Differences of Sphingolipid Molecular Species between the Early Stage of Elongation and the Early Stage of SCW Formation

After initiation (0–2 DPA), the fiber cell starts to elongate, and the rate of elongation gradually increases. At about 10 DPA, the elongation rate reaches a peak (>2 mm/day) [4,5]. After that, the elongation rate began to decrease, and at about 20 DPA, the fiber elongation stops completely [3]. Therefore, at 5 DPA, the elongation rate is increasing while at 15 DPA, the elongation rate is declining. To further compare the differences in sphingolipids between elongation (primary cell wall formation) and SCW formation, we compared the differences in the sphingolipids in the fibers at 5 DPA and 15 DPA (Figure 5A). The results showed that S1P t18:0 and Cer d18:1/16:0 were significantly upregulated at 15 DPA. Eight sphingolipid molecular species, PhytoCer t18:1/24:1, PhytoCer t18:0/18:0, PhytoCer t18:1/h20:0, PhytoCer t18:1/h22:0, PhytoCer t18:1/h24:0, GluCer t18:1/h18:0, GluCer t18:0/h18:0, and PhytoSph, accumulated in high amounts in the elongating fiber cells (Figure 5B). The top five sphingolipid molecular species with the greatest changes were PhytoCer t18:1/h22:0, PhytoCer t18:1/h24:0, PhytoCer t18:1/24:1, PhytoCer t18:1/h20:0, and Cer d18:1/16:0 (Figure 5C). The results also indicated that the PhytoCer molecular species containing unsaturated LCBs and saturated VLCFA were enriched in elongating fiber cells.

### 3.6. The Expression Profile of Genes Involved in Sphingolipid Biosynthesis and Metabolism

To understand the expression profile of the genes involved in sphingolipid biosynthesis and metabolism during the growth and development of fiber cells, we analyzed the expression level of target genes from the FGD database (https://cottonfgd.org/ (accessed on 7 July 2021)) with RT-qPCR. The results showed that the expression levels of most genes were higher at the rapid elongation stage (10 DPA) compared to the SCW synthesis stage (20 DPA), such as *SPTs*, *KSRs*, *LOHs* and *delta8 DESs*; only three detected genes (*SBH1*, *LOH2* and *ACER3-2*) were lower at the elongation stage (Figure 6). Simultaneously, the expression levels of the genes involved in VLCFA biosynthesis and modification were higher at 10 DPA than at 20 DPA (Figure 6). The results indicated that sphingolipid biosynthesis and metabolism were more active in elongating fiber cells. Particularly, the genes involved in LCB unsaturation and hydroxylation as well as in those involved in FA chain elongation and hydroxylation were highly expressed in elongating fiber cells, which might contribute to the high accumulation of unsaturated, tri-hydroxylated Cer-containing hydroxylated VLCFA in elongating fiber cells.

### 3.7. Exogenous Application of Myriocin Inhibited Fiber Elongation

Treatment of the cotton ovules and fiber with FB1, an inhibitor of ceramide synthase, reduced the synthesis of complex sphingolipids but increased the content of simple sphingolipids [33] and inhibited fiber cell elongation. The inhibitory effect of FB1 on fiber elongation may have resulted from both the increase in simple sphingolipids and the decrease in complex sphingolipids. To further illuminate the role of sphingolipids in the growth and development of fiber cells, we treated cotton fibers in vitro with myriocin, which is a potent inhibitor of serine palmitoyltransferase (SPT), the first enzyme of sphingolipid biosynthesis. The results showed that fiber elongation was strikingly suppressed when the ovule was incubated in the medium with 20 μM myriocin (Figure 7), while the repression of fiber elongation was attenuated when the ovule was incubated in the medium with 20 μM myriocin and 10 μM Sph (Figure 7). This indicated that blocking sphingolipid biosynthesis would significantly inhibit the growth of fibers and suggested that sphingolipids play important roles in the growth and development of fiber cells.

## 4. Discussion

Cotton fiber is a seed trichome formed by the polar expansion of the outer integument of the epidermal cells of the ovule. The development process of fiber cells can be divided into four distinct and overlapping periods: initiation, elongation, secondary wall synthesis, and maturation [3,6]. The elongation period and secondary wall synthesis period of fiber cells last for a long time (about 20 days) and are not disrupted by cell division. Therefore, fiber cells are ideal materials for studying cell elongation and secondary cell wall synthesis [3]. Furthermore, these two periods are also the key periods that determine the fiber quality (fiber length and strength) of cotton. The fiber elongation mode is a linear cell growth mode that includes diffusion growth and tip growth [4]. An obvious feature of elongating fibers is the increase in cell length and cell membrane size. A remarkable feature of the secondary cell wall synthesis stage is that the cells stop elongating, and the secondary wall continues to thicken. Fiber cell elongation is a complex biological process that involves many biological processes such as synthesis, transportation, and signal transduction. Sphingolipids are an important component of lipid rafts, a functional region of biomembranes, and are also bioactive molecules [8,38,39,40], and the lipid raft activity of membranes changes significantly during fiber cell development [29]. Therefore, in this study, we characterized the content and composition of sphingolipids in fiber cells at four developmental stages and analyzed the differences in sphingolipid molecule species between the rapid elongation stage (10 DPA) and the secondary cell wall deposition stage (20 DPA) as well as the differences between the early elongation stage (5 DPA) and the early secondary cell wall synthesis stage (15 DPA). The results showed that the sphingolipid molecule species that became significantly enriched during fiber elongation were mainly phytoceramides (trihydroxy LCB-saturated VLCFA). Because little is known about the function of sphingolipids in fiber development, our discussion mainly focused on the possible role of sphingolipids in regulating membrane properties, protein sorting and transportation, and plant hormone crosstalk in fiber cell growth.

Compared to the secondary cell wall synthesis stage, when the membrane size does not increase, the cell membrane size in the elongation stage needs to increase continuously. At the same time, due to the diffusion growth mode, it is also necessary to maintain high membrane stability to resist high turgor. One of the functions of sphingolipids with a high content in elongating fiber cells may be to provide structural components to support the expansion of the membrane. Sphingolipid molecule species enriched in elongating fibers are mainly molecules containing trihydroxy LCB and saturated VLCFA. Studies on mammals have shown that saturated FA can increase the order of FA chain arrangement at the interface of bilayer membranes, thus significantly increasing the stability and phase transition temperature of membranes (such as Cer in skin). On the contrary, unsaturated FA could weaken the regularity of the FA chain arrangement at the interface of bilayer membranes and could reduce the phase transition temperature and order of membranes (such as Cer in brain). Unsaturated FA on the LCB chain affect the Cer interactions. The hydroxylation of FA and LCB and the chain length of FA can increase the stability of the membrane and can finely regulate the physical properties of the membrane [41]. The molecule species containing trihydroxy LCBs and saturated VLCFA were enriched in elongating fiber cells, indicating that the membrane of elongating fiber cells had higher stability. This is also consistent with previous reports that the membrane of fiber cells at the elongation stage had high lipid raft activity (the membrane lipid arrangement is orderly and stable at this stage) [29,30]. However, the high stability of the membrane during fiber cell elongation may be conducive to maintaining high turgor pressure. Ruan et al. reported that the plasmodesmata at the base of fiber cells were closed during the elongation stage and that the turgor pressure in fiber cells increased, which promoted fiber elongation [4,42,43]. The exogenous application of saturated VLCFA such as C24 and C26 FA promoted fiber elongation, and the exogenous application of ACE, an inhibitor of VLCFA synthesis, suppressed fiber elongation [34], which further confirmed that sphingolipids containing VLCFA are required for fiber cell elongation.

Given that fiber cell elongation occurs in a linear cell-growth mode [4], compared to the secondary wall cell synthesis stage (mainly cellulose synthesis), intracellular membrane trafficking is active in the elongation stage. The high accumulation of sphingolipids may also be responsible for the sorting and transportation of intracellular proteins. Previous studies have shown that different sphingolipids species are involved in specific steps of the intracellular trafficking of auxin carriers. Auxin is an important phytohormone, and auxin carriers are localized at the plasma membrane in a polar manner and drive the polar distribution of auxin in plant tissues [44,45,46,47]. A major intracellular sorting station for remodeling the polarity of auxin carriers is the post-Golgi compartment called the trans-Golgi network (TGN) [39]. The TGN is composed of different subgroups or subdomains, which have different protein and lipid components. Golgi secretory vesicles, (SVs)–TGNs, are enriched in hVLCFAs, while clathrin-coated vesicles, (CCVs)–TGNs, are not [48]. At the cellular level, kcs mutants or metazachlor (an inhibitor of VLCFA synthesis) treatment blocked the secretory sorting of the auxin efflux vector PIN2 but not PIN1 at the SVs–TGN [48]. In contrast, loh mutants or FB1 treatment resulted in endocytosis blocking of the auxin efflux vector PIN1 rather than PIN2 [12,48]. Furthermore, reducing the acyl chain length had a strong impact on protein sorting and TGN morphology. The TGN was more expanded and less interconnected with tubules [48]. Compared to SLs with an acyl chain length of 18 carbon atoms, the acyl chain length of 24 carbon atoms formed an interdigitated gel phase and membrane tubules [49]. These studies showed that sphingolipids containing VLCFA are conducive to protein sorting and transportation. During cotton fiber cell elongation, the phytoceramides enriched in VLCFA, especially hVLCFA, may participate in and promote the sorting and transportation of the proteins required for cell elongation. Auxin plays an important role in the development of cotton fiber cells [6,50]. The changes in sphingolipid composition and content may be involved in the transportation and localization of PINs in fiber cells, thus regulating the growth of fiber cells.

Cotton fibers are single cells that arises from the epidermis of the ovule. The connection between the base of fiber cells and other cells is indispensable for the growth of fiber. Plasmodesmata (PD) are membrane channels connecting the PM. The function of the PD is to create the PM, endoplasmic reticulum, and cytoplasmic continuity between cells. The PD allows the active transport of a large number of molecules, such as proteins, transcription factors, or RNA, among cells [51]. Sterols and VLCFA sphingolipids accumulate at the entry of the PD channel. Two GPI anchored proteins, PD callose binding protein 1 (PDCB1) and PD β-glucanase 2 (PDBG2), are involved in callose homeostasis, and their localization depends on sphingolipids [52,53,54,55]. Another study described the role of sphingolipids in PD maturation. The phloem unloading regulator (PLM) protein is predicted to encode an enzyme belonging to the sphingomyelin synthase family [56]. These results proved that sphingolipids play an important role in the formation and regulation of the PD. Recently, a report indicated that saturated LCBs, especially sphinganine (d18:0) and phytosphingosine (t18:0), were enriched in the PD. The t18:0 LCB facilitates the recruitment of PDLP5 at the PD through direct binding to initiate callose deposition, which, in turn, induces PD closure [57]. Ruan et al. reported that the plasmodesmata at the base of cells were closed at the fiber elongation stage and that the turgor pressure in cells increased, which promoted fiber elongation [4,42,43]. In our study, the high content of t18:0 Sph present during the elongation stage of fiber cells (Figure 2A) suggested that t18:0 Sph might play a role in regulating PD closure.

Existing studies have shown that sphingolipids are closely related to plant hormones in plant growth and development. S1P regulates the stomata size during plant hormone abscisic acid (ABA) signaling through the pathway mediated by heterotrimeric G protein signaling [20,21,22,27,58]. In fact, other studies have also shown that there is a close functional relationship between sphingolipids and plant hormone mediated signals, such as PCD. In the process of PCD, the proportion of LCB and CER affects the JA and SA signal pathways [59,60]. The degree of LCB hydroxylation plays an important role in plant growth. The sbh1sbh2 double mutant results in seedling lethality, and the RNAi plant growth of SBH1 and SBH2 is seriously impacted [61]. Furthermore, unsaturated LCBs modified multiple factors involved in ethylene signaling, such as RAF-like kinase constructive triple response 1 (CTR1), ethylene insensitive 2 (EIN2), and ethylene insensitive 3 (EIN3) [62,63]; ethylene also affects sphingolipid synthesis [64]. These studies revealed that sphingolipids play an important role in plant hormone signaling. Many plant hormones, such as brassinosteroids, ethylene, auxin, and gibberellin, play important roles in the growth and development of cotton fiber cells [6,65]. However, the relationship between sphingolipids and plant hormones during fiber development has not been reported, which may be an important topic in the future.

## 5. Conclusions

Cotton fiber cells are an ideal material for studying plant cell elongation, cellulose synthesis, and SCW deposition since cell elongation is not interrupted by cell division and lasts for a long time. In this study, we detected the composition and content of sphingolipids at various stages of fiber cell growth and development and further analyzed the changes in sphingolipid molecular species and their content during fiber elongation and SCW synthesis. The results showed that sphingolipids play an important role in the growth and development of fiber cells. The predominant sphingolipids in fiber are PhytoCer, PhytoCer-OHFA, phyto-GluCer, and GIPC. The phyto-S1P and PhytoCer content changed greatly with fiber development. The Cer molecular species containing tri-hydroxyl, unsaturated LCBs, and hydroxylated saturated VLCFA (C20–C26) accumulated in large amounts during the fiber cell elongation stage, while those containing di-hydroxyl LCB and VLCFA (C20 and C22) accumulated in large amounts during the SCW synthesis stage. Consistently, the genes involved in VLCFA synthesis and hydroxylation, LCB synthesis, LCB hydroxylation and desaturation, and Cer synthesis were highly expressed during the fiber elongation stage. These results indicate that sphingolipids play a role in fiber cell elongation and SCW deposition, which provides a new perspective for further studies on the regulatory mechanism of cotton fiber cell development.

## Figures and Tables

**Figure 1 biomolecules-11-01352-f001:**
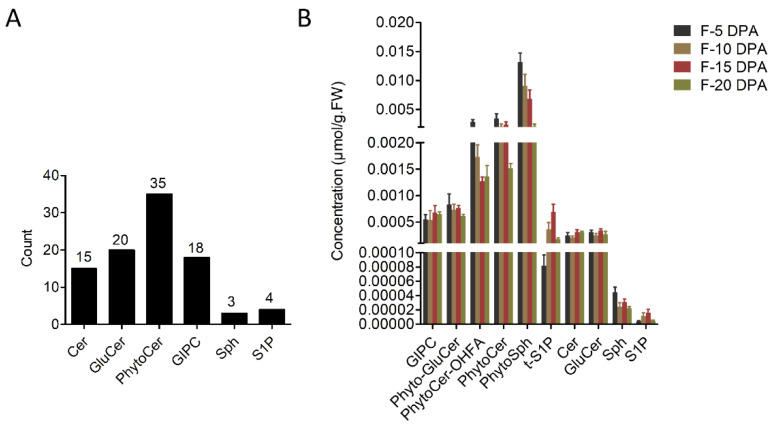
Sphingolipid classes in cotton fiber cells and their alteration trends during fiber growth. (**A**) The number of classes and molecular species of sphingolipids detected in fiber cells. (**B**) The content of sphingolipid classes at four developmental stages of cotton fiber cells. Cer, ceramides; PhytoCer, phytoceramides; PhytoCer-OHFA, phytoceramides with hydroxylated fatty acyls; S1P, sphingosine-1-phosphate; t-S1P, phytosphingosine-1-phosphate; Sph, sphingosines; PhytoSph, phytosphingosines; GluCer, glucosylceramides; Phyto-GluCer, phyto-glucosylceramides; GIPC, glycosyl-inositol-phospho-ceramides. F-5 DPA, F-10 DPA, F-15 DPA, and F-20 DPA represent fiber cells at 5, 10, 15, and 20 DPA (days post-anthesis).

**Figure 2 biomolecules-11-01352-f002:**
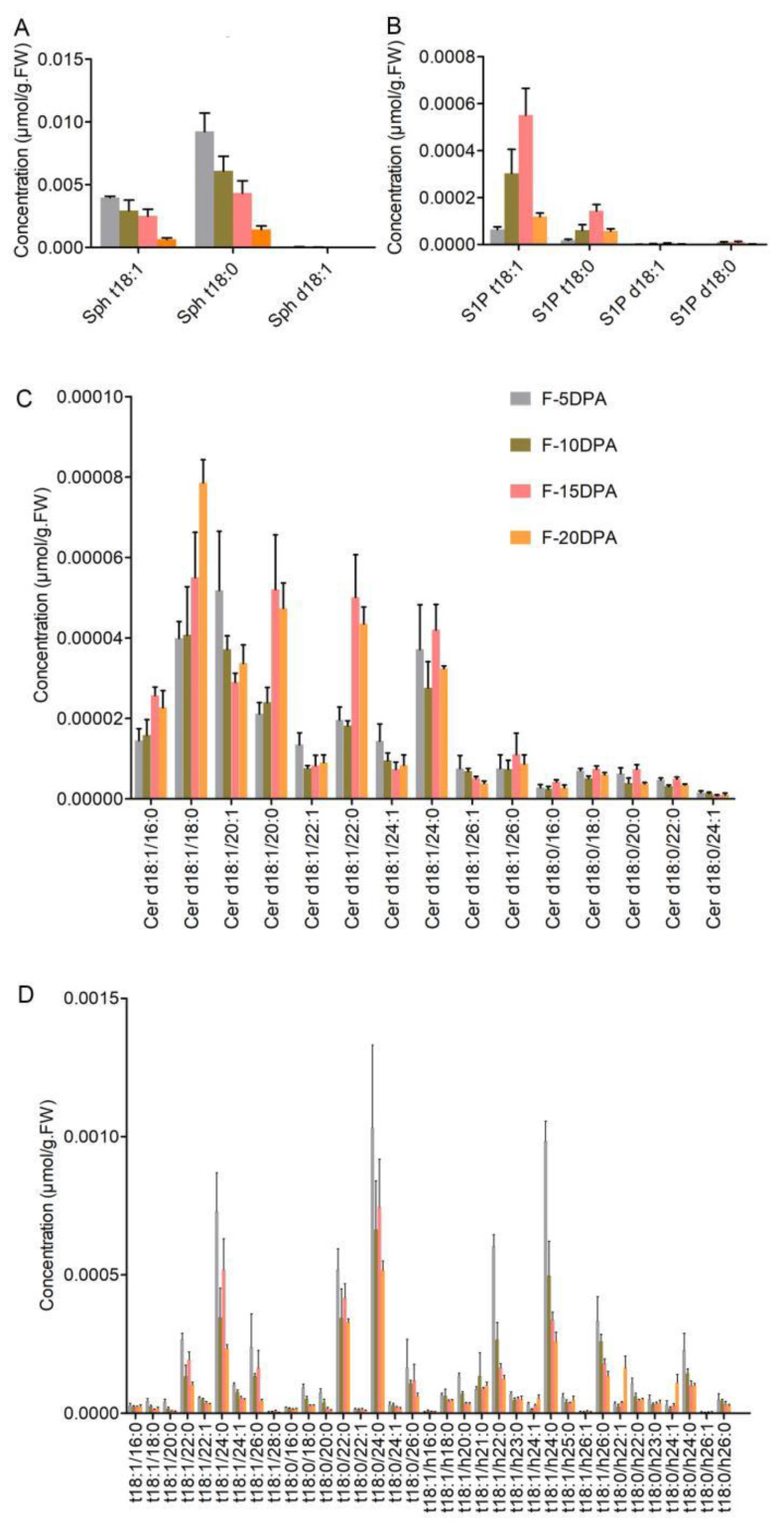
The concentration of simple sphingolipids in cotton fiber cells at four developmental stages. (**A**) The concentration of various molecular species of Sph. (**B**) The concentration of various molecular species of S1P. (**C**) The concentration of various molecular species of Cer. (**D**) The concentration of various molecular species of PhytoCer in cotton fiber cells at four developmental stages. Sph, sphingosines; S1P, sphingosine-1-phosphate; Cer, ceramides. “d18:0/1” indicates that the long-chain bases (LCB) of sphingolipids had w hydroxyl groups (d), 18 carbon atoms, and no or 1 double bond; “t18:0/1” indicates that the LCB had 3 hydroxyl groups (t), 18 carbon atoms, and no or 1 double bond; “16-26:0/1” indicates that the long-chain fatty acid (LCFA) of sphingolipids had 16 to 26 carbon atoms and no or one double bond; and “h16-26:0/1” indicates that the long-chain fatty acid (LCFA) of sphingolipids was a hydroxylated fatty acyl (h) and had 16 to 26 carbon atoms and no or 1 double bond. F-5 DPA, F-10 DPA, F-15 DPA, and F-20 DPA represent fiber cells at 5, 10, 15, and 20 DPA. Each analysis was repeated with three biological replicates.

**Figure 3 biomolecules-11-01352-f003:**
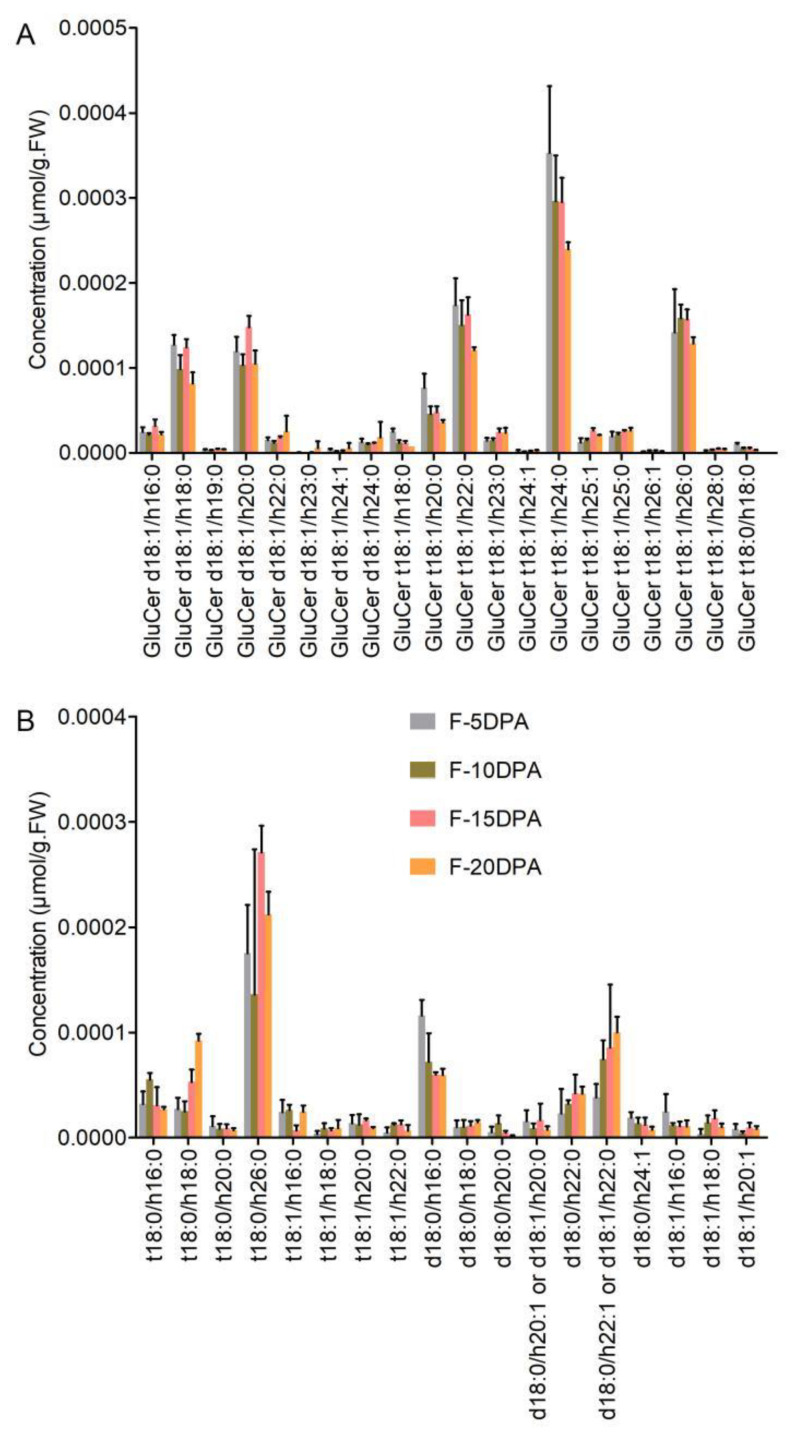
The concentration of complex sphingolipids in cotton fiber cells at four developmental stages. (**A**) The concentration of various molecular species of GluCer. (**B**) The concentration of various molecular species of GIPC in cotton fiber cells at four developmental stages. GluCer, glucosylceramides. “d18:0/1” indicates that the long-chain bases (LCB) of sphingolipids had 2 hydroxyl groups (d), 18 carbon atoms and no or 1 double bond; “t18:0/1” indicates that the LCB had 3 hydroxyl groups (t), 18 carbon atoms, and no or 1 double bond; “16-26:0/1” indicates that the long-chain fatty acid (LCFA) of sphingolipids had 16 to 26 carbon atoms and no or 1 double bond; and “h16-26:0/1” indicates that the long-chain fatty acids (LCFA) of sphingolipids were hydroxylated fatty acyls (h) and had 16 to 26 carbon atoms and no or 1 double bond. F-5 DPA, F-10 DPA, F-15 DPA, and F-20 DPA represent fiber cells at 5, 10, 15, and 20 DPA. Each analysis was repeated with three biological replicates.

**Figure 4 biomolecules-11-01352-f004:**
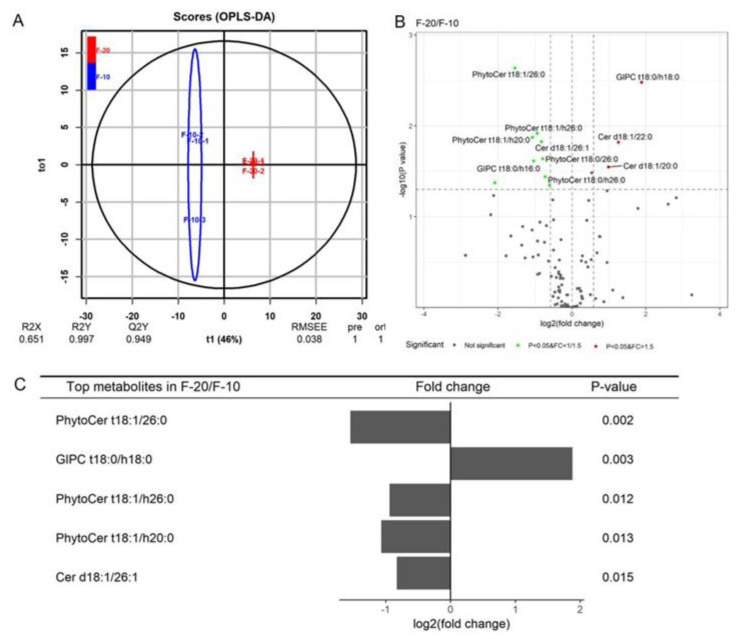
Differences in sphingolipids species in fiber cells between 10 DPA and 20 DPA. (**A**) The OPLS-DA score plot between the 10 DPA group and the 20 DPA group. (**B**) Volcano map of sphingolipid molecular species in fiber cells between 10 DPA and 20 DPA. (**C**) The fold changes of the five top sphingolipid molecular species that changed significantly between 10 DPA and 20 DPA. Cer, ceramides; PhytoCer, phytoceramides; GIPC, glycosyl-inositol-phospho-ceramides. “d18:0/1” indicates that the long-chain bases (LCB) of sphingolipids had 2 hydroxyl groups (d), 18 carbon atoms, and no or 1 double bond; “t18:0/1” indicates that the LCB had 3 hydroxyl groups (t), 18 carbon atoms, and no or 1 double bond; “16-26:0/1” indicates that the long-chain fatty acid (LCFA) of sphingolipids had 16 to 26 carbon atoms and no or 1 double bond; and “h16-26:0” indicates that the long-chain fatty acids (LCFA) of sphingolipids were hydroxylated fatty acyls (h) and had 16 to 26 carbon atoms and no double bond.

**Figure 5 biomolecules-11-01352-f005:**
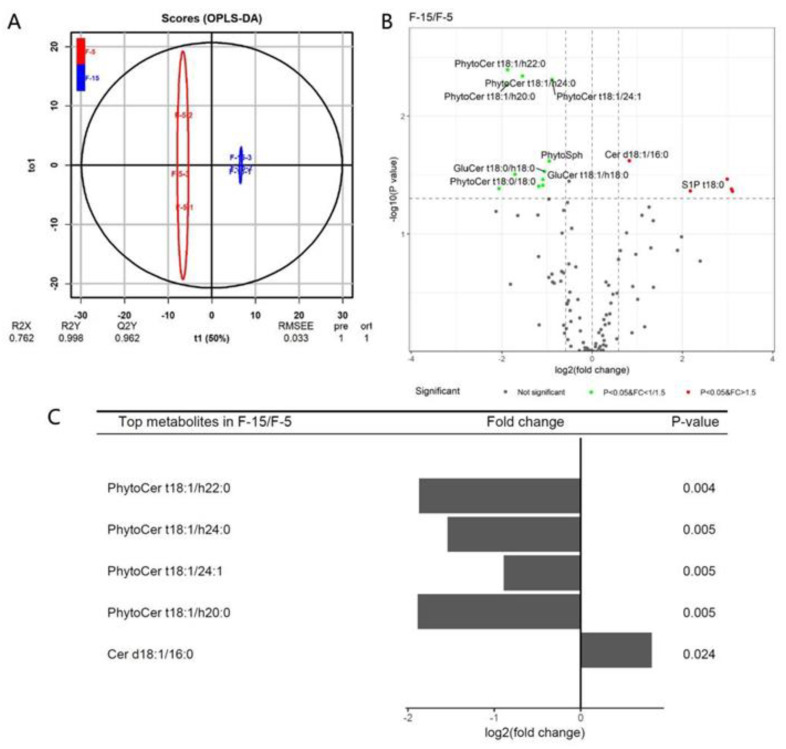
Different sphingolipid molecular species in fiber cells between 5 DPA and 15 DPA. (**A**) OPLS-DA score plot between the 5 DPA group and the 15 DPA group. (**B**) Volcano map of sphingolipid molecular species in fiber cells between 5 DPA and 15 DPA. (**C**) The fold changes of the top five sphingolipid molecular species that significantly changed between 5 DPA and 15 DPA. Cer, ceramides; PhytoCer, phytoceramides; S1P, sphingosine-1-phosphate; PhytoSph, phytosphingosines; GluCer, glucosylceramides. “d18:1” indicates that the long-chain bases (LCB) of sphingolipids had t2 hydroxyl groups (d), 18 carbon atoms, and 1 double bond; “t18:1” indicates that the LCB had 3 hydroxyl groups (t), 18 carbon atoms, and 1 double bond; “16-26:0/1” indicates that the long-chain fatty acid (LCFA) of sphingolipids had 16 to 26 carbon atoms and no or 1 double bond; and “h16-26:0” indicates that the long-chain fatty acids (LCFA) of sphingolipids were hydroxylated fatty acyls (h) and had 16 to 26 carbon atoms and no double bond.

**Figure 6 biomolecules-11-01352-f006:**
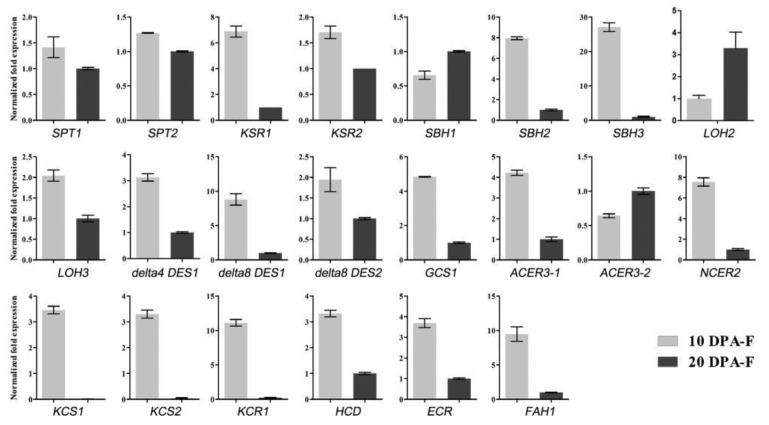
The expression profile of genes involved in sphingolipid biosynthesis and metabolism. *SPT1* and *SPT2*, *serine palmitoyltransfersase*; *KSR1* and *KSR2*, *3-ketosphinganine reductase*; *SBH1*, *SBH2*, and *SBH3*, *LCB C-4 hydroxylase*; *LOH2* and *LOH3*, *ceramide synthase*; *delta4 DES1*, *LCB delta4 desaturase*; *delta8 DES1* and *delta8 DES2*, *LCB delta8 desaturase; GCS1*, *glucosylceramide synthase*; *ACER3*-1 and *ACER3-2*, *alkaline ceramidease*; *NCER2*, *neutral ceramidease*; *KCS1* and *KCS2*, *3-ketoacyl-CoA synthase*; *KCR*, *3-ketoacyl-CoA reductase*; *HCD*, *hydroxyacyl-CoA dehydrase*; *ECR*, *enoyl-CoA reductase*; *FAH1*, *fatty acid 2-hydroxylase*. Three independent RNA isolations were used for cDNA synthesis, and each cDNA sample was subjected to quantitative real-time PCR analysis in triplicate. Error bars represent the SD.

**Figure 7 biomolecules-11-01352-f007:**
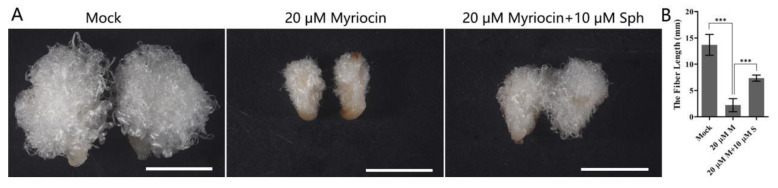
Exogenous application of myriocin and Sph to an in vitro ovule culture system. (**A**) Cotton ovules were collected at 2 DPA and were cultured for 10 days in the in vitro ovule culture system. BT medium adjusted with the amount of DMSO equivalent to that used to dissolve myriocin and sphingosine was used as the mock system. Bar = 5 mm. (**B**) The length the of fibers cultured for 10 days in the ovule culture system. Here, 20 μM M represents the 20 μM myriocin treatment, and 20 μM M + 10 μM S represents the 20 μM myriocin and 10 μM sphingosine treatment. Error bars represent the SD for at least 10 seeds. Triple asterisks indicate statistically significant differences between various treatments, as determined by Student’s *t*-test (*** *p* < 0.001).

## Data Availability

Not applicable.

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
