# Peer review of "Sphingolipid Profile during Cotton Fiber Growth Revealed That a Phytoceramide Containing Hydroxylated and Saturated VLCFA Is Important for Fiber Cell Elongation"

_biomolecules, 2021, doi:10.3390/biom11091352_

Round 1
Reviewer 1 Report
In this manuscript, the authors analyzed the role of sphingolipids during cotton fiber elongation by analyzing the sphingolipid species and amount during cotton fiber elongation by UHPLC-MS/MS, expression of genes involved in sphingolipid biosynthesis, and finally the effect of sphingolipid biosynthesis inhibitor on cotton fiber development. Authors found that some sphingolipid species showed changes in their amounts during cotton fiber development and inhibiting sphingolipid biosynthesis significantly suppresses cotton fiber length.
While those analyses do suggest that there is some correlation between cotton fiber development and sphingolipid, they are not sufficient to conclude that sphingolipid is important for cotton fiber elongation, which is the focus of this manuscript. Moreover, the presentation and writing of this manuscript are very poor, and it is very hard for readers who are not specialists in sphingolipid to follow the manuscript. Other specific comments are listed below.
Major comment
- Based on the myriocin treatment and myriocin+Sph treatment on cotton fiber, the authors concluded that sphingolipid is important for fiber cell elongation. However, with the data shown in Figure 7, it is not clear that if this inhibitor treatment indeed inhibits cell elongation or it also inhibits cell proliferation. Authors must include cytological data to show that cell elongation but not cell proliferation is affected in this experiment. Otherwise, if there are previous papers showing that there is only cell elongation taking place but no cell proliferation at this developmental stage when cotton fiber was treated, authors must refer to such papers and include sufficient information in the manuscript.
Minor comments
- Abbreviations must be defined in the first place appeared in the manuscript. Particularly in the abstract authors used undefined abbreviations, which makes readers impossible to understand what is described.
- Authors must include figures for sphingolipids' chemical structures and biosynthetic pathways and include basic information of sphingolipid in the manuscript. For biosynthetic pathways, authors must include information regarding which proteins mediate the reactions, so readers can understand the roles of genes analyzed in qRT-PCR (Figure 6).
- While the main data of this manuscript is UHPLC-MS/MS analysis, methods to obtain those data were omitted in the Materials and Methods section. Authors must give detailed protocol in the Methods section, not just referring to previous papers.
- While authors described that they identified 95 molecular species for sphingolipid in cotton fiber, there is no list for the identified species. Authors must include the list.
- It is not clear why the authors used different categorizations for Figures 1A and 1B. Authors must explain why they used different categorizations.
- In Figure 1B, the broken axis covers the top part of many bars and makes the figure hard to see for readers. Authors must revise this figure.
- In p2 L156, the authors explained “The content of phyto-S1P was lower in 5 DPA”, but in Figure 1B there is no phyto-S1P. Authors must use the unified term to indicate the same molecules throughout the manuscript.
- Figure 2, explain the difference between t18:1 and d18:1.
- p5 L203-206. “indicate” is a too strong word for those data. The authors only showed that there is a correlation between cotton fiber development and some of sphingolipid species, which does not necessarily mean that there is a causal relationship.
- In Figures 4A and 5A, authors must explicitly explain what the OPLS-DA score is, how those scores are obtained, and what the data mean.
Author Response
Dear Reviewer:
On behalf of my co-authors, we appreciate you very much for their positive and constructive comments and suggestions on our manuscript entitled “Sphingolipid profile during cotton fiber growth revealed that a phytoceramide containing hydroxylated and saturated VLCFA is important for fiber cell elongation.”(biomolecules-1362178). We have studied your comments carefully and have made revision in the paper. We have tried our best to revise our manuscript according to the comments. Attached please find the revised version, which we would like to submit for your kind consideration. Here are our point-by-point response:
Major comment
- Based on the myriocin treatment and myriocin+Sph treatment on cotton fiber, the authors concluded that sphingolipid is important for fiber cell elongation. However, with the data shown in Figure 7, it is not clear that if this inhibitor treatment indeed inhibits cell elongation or it also inhibits cell proliferation. Authors must include cytological data to show that cell elongation but not cell proliferation is affected in this experiment. Otherwise, if there are previous papers showing that there is only cell elongation taking place but no cell proliferation at this developmental stage when cotton fiber was treated, authors must refer to such papers and include sufficient information in the manuscript.
Answer: Thank you for your valuable comment. We have described in the introduction that the cotton fiber is a single cell. “Cotton fiber is a single-celled fiber formed by polar elongation and secondary cell wall (SCW) thickening of the ovule epidermal cell, which is considered as an ideal material for studying cell elongation, cellulose synthe-sis and secondary wall deposition”
Minor comments
- Abbreviations must be defined in the first place appeared in the manuscript. Particularly in the abstract authors used undefined abbreviations, which makes readers impossible to understand what is described.
Answer: Thank you very much for your suggestion. We have explained these acronyms.
- Authors must include figures for sphingolipids' chemical structures and biosynthetic pathways and include basic information of sphingolipid in the manuscript. For biosynthetic pathways, authors must include information regarding which proteins mediate the reactions, so readers can understand the roles of genes analyzed in qRT-PCR (Figure 6).
Answer: Thank you for your careful comment. We have added the biosynthetic pathways for sphingolipid synthesis in the introduction section.
- While the main data of this manuscript is UHPLC-MS/MS analysis, methods to obtain those data were omitted in the Materials and Methods section. Authors must give detailed protocol in the Methods section, not just referring to previous papers.
Answer: Thank you for your good suggestion. We have given detailed protocol about UHPLC-MS/MS analysis in the Methods section.
- While authors described that they identified 95 molecular species for sphingolipid in cotton fiber, there is no list for the identified species. Authors must include the list.
Answer: Thank you for your good suggestion. We have added the list for the identified 95 molecular species of sphingolipids in table S2.
- It is not clear why the authors used different categorizations for Figures 1A and 1B. Authors must explain why they used different categorizations.
Answer: Figure 1A is for “The number of classes and molecular species of sphingolipids” and Figure 1B is for “The content of sphingolipid classes”.
- In Figure 1B, the broken axis covers the top part of many bars and makes the figure hard to see for readers. Authors must revise this figure.
Answer: Thank you for your careful comment. We have revised this figure.
- In p2 L156, the authors explained “The content of phyto-S1P was lower in 5 DPA”, but in Figure 1B there is no phyto-S1P. Authors must use the unified term to indicate the same molecules throughout the manuscript.
Answer: Thank you for your good suggestion. We accepted and corrected.
- Figure 2, explain the difference between t18:1 and d18:1.
Answer: Accepted and added.
- p5 L203-206. “indicate” is a too strong word for those data. The authors only showed that there is a correlation between cotton fiber development and some of sphingolipid species, which does not necessarily mean that there is a causal relationship.
Answer: Accepted and corrected.
- In Figures 4A and 5A, authors must explicitly explain what the OPLS-DA score is, how those scores are obtained, and what the data mean.
Answer: Accepted and added.
We would like to express our great appreciation to you for comments on our paper. Looking forward to hearing from you.
Thank you and best regards.
Yours sincerely,
Ming Luo
Reviewer 2 Report
Review of the MSC titled „Sphingolipid profile during cotton fiber growth revealed the phytoCeramide containing hydroxylated and saturated VLCFA is important for fiber cell elongation” by Chen et al. submitted to BIOMOLECULES.
The topic of these investigations, namely establishing a relationship between the sphingolipid profile and developmental stages of the cotton fiber, is important both from the perspective of involvement of biomolecules in plant cell development and for improvement of cotton, the major crop used for textile production. Although the presented results are important, in my opinion, the MSC has significant flaws that need to be corrected.
Major comments:
- The formation of cotton fiber, the process fundamental for this MSC, needs to be presented in a clear way. First, indeed the common name “cotton fiber” is often used in literature but when introducing the object of their investigation the Authors need to explain that this is not the fiber in a morphological (anatomical) sense but a trichome formed by the seed coat epidermis (see e.g. citation [7] of this MSC). Only after such introduction the term “cotton fiber” can be used. Second, when describing the process of cotton fiber formation it needs to be mentioned that the fiber elongates due to a combination of tip and diffuse growth (as explained by [4] cited in this MSC). This is important for the interpretation of the results because if only tip growth took place only membranes located in the tip region of fiber cell would be important for growth, and their contribution in samples analyzed in this study would be very low or even negligible. Moreover, what do the authors mean by “polar elongation”? This is a rather awkward term. Third, it has to be clearly stated that the stages of cell elongation and deposition of secondary cell wall do not overlap (by definition the secondary wall is formed after growth cessation).
The Discussion is in my opinion quite poorly written and needs much improvement. First, some conclusions are drawn already in the Results section, and lots of them are not enough supported by the presented data (e.g. the statements in lines 188-189 or 201-204 – there is no support that these are direct effects). Second, several paragraphs of Discussion are simply repetitions from Results. After these repetitions, some reports concerning other species/cell types, mainly arabidopsis, are referred to but with no clear connection to the presented original results. An important problem, which in my opinion needs to be addressed in the Discussion is the implications of the major difference between the elongation stage and secondary wall deposition stage. Namely, in the elongation stage, unlike in the SCW deposition stage, there is a significant increase in the plasma membrane surface. At the same time, new wall material (non-cellulosic) is delivered to the surface in membrane vesicles. How is this membrane expansion accomplished by a cell? What is the contribution of exo- and endocytosis in the process (maybe membrane expansion and exo- and endocytosis affect the membrane composition)? Is there a way to distinguish between the simple requirement of sphingolipid synthesis for membrane surface expansion and their direct involvement in fiber growth regulation? I understand that these are not topics of the MSC but the questions need to be addressed in the Discussion.
Finally, the English language requires major corrections and it is not the reviewer’s task to perform them.
Minor comments in order of appearance:
Abstract – acronyms (e.g. SCW, LCB, VLCFA) are not explained and the reader is not expected to search for these explanations in the main text. It requires the Editor decision whether acronyms of sphingolipid species names can be used in the abstract
Mutant names should be in Italic, the same as Latin species names.
There is some discrepancy between acronyms used in Fig. 1 and description in the main text (lines 145-163).
In figure legends, the phrase “fiber cell (without ovule)” is often used. This is very misleading as it suggests that a fiber includes the ovule, which obviously is not the case.
What do the Authors mean by “secondary wall synthesis lasts about 20-25 days” (lines 243-244) or a similar statement on elongation period (line 242)? Should it not rather be “secondary wall synthesis lasts for 5 days, i.e. from 20 to 25 DPA”? This is confusing.
Statistical methods are not explained in enough detail in the Material & Methods section nor in the figure legends or Results. What is the statistics “FC” referred to in lines 244-256? What are the plots shown in Figs 4-5 (please note that the OPLS-DA method is not even mentioned in the text)?
Figure 7 – I could not find any explanation of how the fiber length was measured. Another question: is the ovule size affected by the treatment? If yes, can the Authors be sure that the observed effects on fiber length are direct?
Line 419 – it is suggested that the compounds under consideration “promote cell to induce PCD and enter the SCW synthesis”. This does not make sense because only the alive cell can deposit new wall layers.
Author Response
Dear Reviewer:
On behalf of my co-authors, we appreciate you very much for their positive and constructive comments and suggestions on our manuscript entitled “Sphingolipid profile during cotton fiber growth revealed that a phytoceramide containing hydroxylated and saturated VLCFA is important for fiber cell elongation.”(biomolecules-1362178). We have studied your comments carefully and have made revision in the paper. We have tried our best to revise our manuscript according to the comments. Attached please find the revised version, which we would like to submit for your kind consideration. Here are our point-by-point response:
Major comments:
- The formation of cotton fiber, the process fundamental for this MSC, needs to be presented in a clear way. First, indeed the common name “cotton fiber” is often used in literature but when introducing the object of their investigation the Authors need to explain that this is not the fiber in a morphological (anatomical) sense but a trichome formed by the seed coat epidermis (see e.g. citation [7] of this MSC). Only after such introduction the term “cotton fiber” can be used. Second, when describing the process of cotton fiber formation it needs to be mentioned that the fiber elongates due to a combination of tip and diffuse growth (as explained by [4] cited in this MSC). This is important for the interpretation of the results because if only tip growth took place only membranes located in the tip region of fiber cell would be important for growth, and their contribution in samples analyzed in this study would be very low or even negligible. Moreover, what do the authors mean by “polar elongation”? This is a rather awkward term. Third, it has to be clearly stated that the stages of cell elongation and deposition of secondary cell wall do not overlap (by definition the secondary wall is formed after growth cessation).
Answer: Thank you for your valuable suggestions. We have rewrote the first sentence of abstract as “Cotton fiber is a single-celled seed trichrome that arises from the epidermis of ovule outer integument”. Moreover, We have changed “polar elongation” to “polar expansion”.
- The Discussion is in my opinion quite poorly written and needs much improvement. First, some conclusions are drawn already in the Results section, and lots of them are not enough supported by the presented data (e.g. the statements in lines 188-189 or 201-204 – there is no support that these are direct effects). Second, several paragraphs of Discussion are simply repetitions from Results. After these repetitions, some reports concerning other species/cell types, mainly arabidopsis, are referred to but with no clear connection to the presented original results. An important problem, which in my opinion needs to be addressed in the Discussion is the implications of the major difference between the elongation stage and secondary wall deposition stage. Namely, in the elongation stage, unlike in the SCW deposition stage, there is a significant increase in the plasma membrane surface. At the same time, new wall material (non-cellulosic) is delivered to the surface in membrane vesicles. How is this membrane expansion accomplished by a cell? What is the contribution of exo- and endocytosis in the process (maybe membrane expansion and exo- and endocytosis affect the membrane composition)? Is there a way to distinguish between the simple requirement of sphingolipid synthesis for membrane surface expansion and their direct involvement in fiber growth regulation? I understand that these are not topics of the MSC but the questions need to be addressed in the Discussion.
Answer: Accepted and we have rewrote the discussion section.
- Finally, the English language requires major corrections and it is not the reviewer’s task to perform them.
Answer: Thank you for your advice. This article has been modified by the language company in terms of language, and there are relevant certificates in the attachment. We have carefully checked our manuscript.
Minor comments in order of appearance:
- Abstract – acronyms (e.g. SCW, LCB, VLCFA) are not explained and the reader is not expected to search for these explanations in the main text. It requires the Editor decision whether acronyms of sphingolipid species names can be used in the abstract
Answer: Thank you for your careful comment. We have explained these acronyms.
- Mutant names should be in Italic, the same as Latin species names.
Answer: Thank you for your advice. We have changed these word into the correct form.
- There is some discrepancy between acronyms used in Fig. 1 and description in the main text (lines 145-163).
Answer: Thank you very much for your suggestion. We have explained “phyto-S1P”. t-S1P is an acronym for phyto-S1P.
- In figure legends, the phrase “fiber cell (without ovule)” is often used. This is very misleading as it suggests that a fiber includes the ovule, which obviously is not the case.
Answer: Thank you for your careful comment. We have removed “without ovule”.
- What do the Authors mean by “secondary wall synthesis lasts about 20-25 days” (lines 243-244) or a similar statement on elongation period (line 242)? Should it not rather be “secondary wall synthesis lasts for 5 days, i.e. from 20 to 25 DPA”? This is confusing.
Answer: Thank you for your suggestion, we have changed this sentences to “The period of secondary wall synthesis lasts about 20-25 days (from ~20 DPA to ~45 DPA).”
- Statistical methods are not explained in enough detail in the Material & Methods section nor in the figure legends or Results. What is the statistics “FC” referred to in lines 244-256? What are the plots shown in Figs 4-5 (please note that the OPLS-DA method is not even mentioned in the text)?
Answer: Accepted and added.
- Figure 7 – I could not find any explanation of how the fiber length was measured. Another question: is the ovule size affected by the treatment? If yes, can the Authors be sure that the observed effects on fiber length are direct?
Answer: Accepted and added the measurement methods for fiber length.
- Line 419 – it is suggested that the compounds under consideration “promote cell to induce PCD and enter the SCW synthesis”. This does not make sense because only the alive cell can deposit new wall layers.
Answer: We have rewrote the discussion section.
We would like to express our great appreciation to you for comments on our paper. Looking forward to hearing from you.
Thank you and best regards.
Yours sincerely,
Ming Luo